# The peripheral blood mononuclear cells preparation and the hematology of *Varanus salvator*

Jitkamol Thanasak[1]*, Tawewan Tansatit[2], Jarupha Taowan[3], Napawan Hirunwiroj[1], Sujit Chitthichanonte[1], Teetat Wongmack[1]

**1** Faculty of Veterinary Science, Department of Clinical Sciences and Public Health, Mahidol University, Salaya, Phutthamonthon, Nakhon Pathom, Thailand, **2** Faculty of Veterinary Science, Department of Pre-clinic and Applied Animal Science, Mahidol University, Salaya, Phutthamonthon, Nakhon Pathom, Thailand, **3** Faculty of Veterinary Science, The Monitoring and Surveillance Center for Zoonotic Diseases in Wildlife and Exotic Animals, Mahidol University, Salaya, Phutthamonthon, Nakhon Pathom, Thailand

* jitkamol.tha@mahidol.ac.th

**Data Availability Statement:** All relevant data are within the paper.

**Funding:** The authors received no specific funding for this work.

## Abstract

The objectives of this study were to evaluate the proper anticoagulants coated in blood-collecting tube for the peripheral blood mononuclear cells (PBMCs) isolation and to evaluate the proper culture temperature for the *Varanus salvator*'s PBMCs, in addition, the hematological characteristics also reported. The heparin treated blood (n = 10) and EDTA treated blood (n = 10) from *Varanus salvator* were obtained for PBMCs evaluation. The PBMCs obtained from the heparin treated blood was significantly higher than that of EDTA treated blood during the culture period ($P < 0.05$) indicated heparin would be more appropriated anticoagulant for blood collection. The PBMCs cultured under 37˚C and 27˚C were not significantly difference on first three days but 37˚C showed significantly higher effect in the following days ($P < 0.05$) indicated both temperatures can be used which 37˚C should be an optimal for PBMCs preparation. The peripheral blood cells of *Varanus salvator* (n = 49) were analyzed for hematological profiles and characteristics which the number of erythrocytes $1.19 \pm 0.04 \times 10^{12}$/L ($1.17–1.35 \times 10^{12}$/L) and WBC $2.41 \pm 0.13 \times 10^{9}$/L ($2.29–2.81 \times 10^{9}$/L) with absolute differential count of heterophils $0.92 \pm 0.02 \times 10^{9}$/L ($0.87–0.95 \times 10^{9}$/L), lymphocytes $1.17 \pm 0.01 \times 10^{9}$/L ($1.15–1.23 \times 10^{9}$/L), azurophils $0.40 \pm 0.01 \times 10^{9}$/L ($0.37–0.42 \times 10^{9}$/L), basophils $0.000 \pm 0.001 \times 10^{9}$/L ($0.004–0.011 \times 10^{9}$/L) and monocytes $0.027 \pm 0.002 \times 10^{9}$/L ($0.028–0.039 \times 10^{9}$/L). These results would play an important role on the cell immunological studies of the *Varanus salvator* in the future.

## Introduction

The monitor lizards or Varanids (*Varanus spp.*) are animals in class Reptilian, order Squamata, family Varanidae [1]. More than fifty to sixty Varanids distribute around the world with the different in habitat and environment [2,3]. In Thailand, there are 5 Varanid species have been reported including Asian water monitor (*Varanus salvator macromaculatus*), Rough Necked

**Competing interests:** The authors have declared that no competing interests exist.

Monitor (*Varanus rudicollis*), Clouded monitor (*Varanus bengalensis nebulosis*), Dumeril's Monitor (*Varanus dumerillii*) and Black monitor (*Varanus salvator komaini*) [3]. All Thai Varanids are under the wildlife protection according to the list of Thailand protected species, since their population were decreased due to the loss of habitat, hunting for consumption or leather products [4] which the Asian water monitor is the most threatened species due to their life is closer to humans habitat and agriculture [5,6]. As well as in Thailand, the Asian water monitor is extremely widespread throughout Southeast Asia which also can be found in Bangladesh, Sri Lanka, India, Indochina and Indonesia [2]. It is an important scavenger which decomposed carcass and control the excess animal population. Some evidences showed that the water monitor could control the population of common suckers in Sri Lanka [7], and the population of fish in Thailand [6].

In spite of Thai Varanids which under Thai's wildlife protection, other become popular as the exotice pets such as Savannah monitor, Argus monitor, Crocodile Monitor, Mangrove monitor, Nile monitor, Yellow-headed monitor and so on. The Varanid's hematology and immunology are become necessary for veterinary practitioners, however, the information is less. The limited information from other reptiles such as snake, lizard, turtle and crocodile have been applied [8]. The hematology of reptiles has been more investigated both in blood cells and blood chemistry [9], however, there was a rarely reported in the water monitor [10]. Focusing on immunological purpose, reptile's leukocytes consist of heterophil, lymphocyte, eosinophil, monocyte, azurophil and basophil [9]. The populations of leukocytes in reptiles are species difference, Iguana (*Iguana iguana*) and Monkey-tail skink (*Corucia zebrata*) are rich in heterophil [11,12] while krait snake (*Bungarus fasciatus*) has plenty of azurophil [13]. In addition, the most population of leukocytes in king cobra (*Ophiophagus hannah*), cobra (*Naja kaouthia*) Tegu (*Tupinambis spp.*), Savannah monitor (*Varanus exanthematicus*) as well as Asian water monitor (*Varanus salvator*) are lymphocytes [10,14,15]. There were some evidences in snake, lizard, turtle and crocodile which immune responsiveness depend on body temperature, since a number of lymphocytes were elevated at an appropriated temperature [9,16]. Moreover, lymphocyte related tissues and organ also presented better activities at suitable temperature [17].

The Peripheral blood mononuclear cells (PBMCs), are subpopulation of leukocytes which play a role in cell mediated immunology. They consist of lymphocytes and monocytes in endothermic animals [18] and include azurophils in ectothermic animals [19–21]. In mammals, PBMCs are required for many cellular immune function assays [22,23], however, very rare studies in ectothermic animals' PBMCs have been performed. The preparation of blood PBMCs for cell culture is the first important step for immune cell studies. The principle of PBMCs isolation techniques are well recognized in many laboratories. The appropriated protocols have been set for many species, however, most of them are for mammal's PBMCs isolation [19–22,24]. Since, the reptiles are ectothermic animals which have variety of blood cell population, the appropriated protocols are necessary. To collect the blood for PBMCs purification, the previous reports used either heparin [19,25,26] or ethylenediamine tetraacetic acid (EDTA) [20,27] as the anticoagulants. The anticoagulants that could be the best choice for PBMC extraction are still unclear. Thus, a suitable anticoagulant for Varanid's blood collection for PBMCs assay should be investigated.

The technical standardized for mammalian PBMCs isolation has been applied to non-mammalian blood such as fish [28,29], turtles [20,25,26] and Tuatara [19]. A few adjustments on the same principle can be applied to separate the PBMCs in non-mammalian blood, however, a critical temperature for PBMCs culture is difference. The 5% $CO_2$ incubator always subject to be set at 37˚C for mammalian PBMCs [22,23], however, a wide variety temperature ranges, 20–28˚C, have been reported for non-mammalian PBMCs [19,27,28]. The Asian water

monitor is a poikilothermic animal. They live under the temperature of 27–37˚C in range, which the most activities appears when the body temperature are 30.4–31˚C [30]. The body temperature depends on two factors, internal (endothermic) and external factors (ectothermic) which only 2˚C of body temperature can be adjusted by an internal factor, while the external factor plays more role [31]. The body temperature of water monitor have an important direct impact to metabolism rate, oxygen consumption [31], acid-base balance [32], size of red blood cells [33], reproductive system and particularly immunity [8,16]. Thus, this information can be condsidered as a reference for water monitor's PBMCs culture.

Therefore, the aims of this study are 1) to evaluate the proper anticoagulants, EDTA or heparin, for the collecting blood of Asian water monitor for the PBMCs isolation methodologies, 2) to evaluate the proper temperature and the time optimize for Asian water monitor's PBMCs culture survival, and 3) to also analyze the hematological characteristics of Asian water monitor for a clinical reference. These evidences would be a *Varanus salvator* PBMCs preparation protocol for future monitoring applications and will be a starting point of monitor lizard's immunological study which should do a good impact on further conservation and clinical trials.

## Materials and methods

### Animals and sampling

The animals obtained in this experiment are *Varanus salvator* from Kao-Shon wildlife center in Ratchaburi province of Thailand. The adult males and females which more than 3 kilogram of body weight and more than 100 centimeters in a length of snout to anus were randomly obatained for blood collection. All animals did not have any clinical sign including injuries, wounds and emaciation and have a good body condition status. The study was approved by the Faculty of Veterinary Science-Animal Care and Use Committee (FVS-ACUC), Mahidol University, Thailand. (Protocol No. MUVS-2016-08-29). The methods were carried out in accordance with the approved guidelines.

### Preparation of peripheral blood mononuclear cells

Peripheral blood (n = 10) was drawn from the caudal tail vein of *Varanus salvator* using a 18-gauge needle into heparin-treated tubes (4 ml.) and EDTA-treated tubes (4 ml.). All samples were subjected to be isolated for peripheral blood mononuclear cells (PBMCs) in laboratory within 12 hours. The heparin peripheral blood or the EDTA peripheral blood was dilution (1:2) in RPMI-1640 supplemented with 2% fetal bovine serum (FBS), heparin (5 IU/ml), penicillin G (200 IU/ml) and streptomycin (200 μg/ml). Subsequently, the dilution was density gradient centrifuged on HiSep$^{TM}$ LSM 1077 (d = 1.0770± 0.0010 g/ml). The PBMCs were collected from the interphase, and then washed in RPMI-1640 twice. The final cell pellets were resuspended to $2 \times 10^6$ cell/ml in a culture medium which consisted Iscoves supplemented with penicillin G (50 IU/ml), streptomycin (50 μg/ml), L-glutamine (2 mM), ß2-mercapto-ethanol ($2 \times 10^{-5}$ M) and 10% fetal bovine serum (FBS) and then put into round bottom 96-well plates. Each sample was incubated at 27˚C and 37˚C in 5% $CO_2$ humidified atmosphere. The PBMCs on day 1 to 10 were monitor cell lifespan by Trypan blue stainning for viable count in hemocytometer at inverted microscope (10X).

### Haematology

Peripheral blood (n = 49) was obtained from the caudal tail vein of *Varanus salvator* using a 18-gauge needle and was collected on EDTA-treated tubes (4 ml.) for hematologic studies. Red

blood cells and white blood cells counts were analyzed by using Neubauer hemocytometer and Natt-Herrick solution as the diluent [34]. The packed cell volume (PCV) was determined by micro-hematocrit method using capillary tubes. Blood smears were performed, air-dried, fixed in ethanol, and stained in Modified Wright-Giemsa. They were observed under a light microscope (Nikon ECLIPSE E 200-Japan). Two hundred leukocytes were counted for each blood smear and identified following criterias proposed by Stacy et al. [9], Arikan and Cicek [35], Sykes and Klaphake [36].

## Statistical analysis

According to the experimental animals of this study, *Varanus salvator* is wild animal which under the wildlife protection according to the list of Thailand protected species. Thus, the Varanids which were different in sex, ages and size were randomized for the sample collection, 10 Varanids for PBMC experiment and 49 Varanids for hematological experiment. For the PBMC experiment, blood sample drawn from each animal was divided into two treatments of anticoagulant factors (heparin and EDTA) which later used for PBMCs' culture study within two culture temperature treatments (37 and 27˚C). Therefore, the non-parametric statistical analysis should be appropriated for this experimental design which the central tendency of dependent variables measured as median values were presented. The differences in the median values of PBMCs in each treatment according to the culture day effect were compared using the Friedman Test method. The comparison of the difference of median values of PBMCs in each factor, anticoagulants effect and culture temperature effect, followed Wilcoxon Test method. All statistical analyzes were performed using SPSS statistics for Windows version 18.0. The comparison of the median differences across treatments determined statistical significance for the Chi-square test of the dependent variables at the significance level at $P < 0.05$.

The hematological analysis of the number and size of Varanids' peripheral blood cells, were presented by median, standard error and 95% confidence interval. The comparison between the hepatozoon infection and non-infection of dependent hematological variables was determined using Wilcoxon Test method with statistical significance level at $P < 0.05$. The statistical analyzes were performed using SPSS statistics for Windows version 18.0.

## Results and discussion

### The study of peripheral blood mononuclear cells (PBMCs)

The results of *Varanus salvator*'s PBMCs isolated from heparin or EDTA blood which cultured under either 37˚C or 27˚C have showed the statistical difference among 4 experimental factors (Table 1). The volume of most cultured PBMCs isolated from heparin blood showed no statistical difference between the culture temperature, 37˚C or 27˚C. Some of the PBMCs culture at 27˚C seem to be lower on day 6 and 8 ($P < 0.05$). The cultured PBMCs isolated from EDTA blood also showed no statistical difference between 37˚C or 27˚C, on day 1, 2, 3, 7 and 9, however, lower amount on day 6, 8 and 10 were presented by 27˚C cultured ($P < 0.05$). The statistical difference appeared to be evident with the anticoagulant factors (Table 1), which can be presented more clearly in Table 2. The amount of cultured PBMC obtained from the heparin treated blood was significantly higher than that of EDTA treated blood in all days ($P < 0.05$) (Table 2). These results indicated that both anticoagulants can be used for this matter, which heparin presented more efficiency than EDTA. The EDTA has the property to bind to calcium, which is necessary for the blood clotting process. EDTA blood was suitable for hematological examination because EDTA will maintain the good condition of the granules and the shape of blood cells [37]. The heparin blood had recommended for chemical tests such as electrolytes, sugars, lipids, and enzymes due to the ability of anti-thrombin activity which is a coagulation

**Table 1. Effect of either blood anticoagulants or culture temperatures on the peripheral blood mononuclear cells (PBMCs) isolation in *Varanus salvator*.**

| Items | Heparin | | EDTA | | P-value |
|---|---|---|---|---|---|
| | 37˚C | 27˚C | 37˚C | 27˚C | |
| | ————————PBMC, x10^5 cells/mlr———————— | | | | |
| Day 0 | 2.00 | 2.00 | 2.00 | 2.00 | ND |
| Day 1 | 7.10[a] | 7.75[a] | 4.40[b] | 4.75[b] | 0.02 |
| Day 2 | 5.95[a] | 6.00[a] | 3.55[ab] | 4.04[b] | 0.01 |
| Day 3 | 4.35[a] | 3.95[a] | 3.15[ab] | 3.00[b] | 0.00 |
| Day 6 | 4.50[a] | 3.50[b] | 2.93[b] | 2.30[c] | 0.00 |
| Day 7 | 4.00[a] | 2.90[a] | 2.40[ab] | 2.18[b] | 0.00 |
| Day 8 | 3.85[a] | 2.75[b] | 2.19[b] | 1.75[c] | 0.00 |
| Day 9 | 3.25[a] | 2.35[a] | 2.00[ab] | 1.55[b] | 0.00 |
| Day 10 | 2.95[a] | 2.15[a] | 1.65[b] | 1.30[c] | 0.00 |

[a,b,c] Median values with different superscript letters within the same row were statistically significant (Friedman test, $P < 0.05$).

ND = not determined.

factor [38]. Thus, the PBMCs pattern in this study indicated that heparin blood would appropriate for further study of Varanus's PBMCs both in activities and function.

Considering to the culture temperature, 37˚C or 27˚C on PBMCs, there seems to have less effect on each PBMCs derived from heparin blood or EDTA blood (Table 1), which can be showed more clearly in Table 3. There was no statistical difference between PBMCs that cultured in both temperatures, 37˚C and 27˚C, on first 3 days of cell culture, however, the 37˚C showed significant higher effect to the rest ($P < 0.05$) (Table 3). These results are in contrast with a previous report in tuatara, which an appropriate temperature for tuatara's PBMCs culture was 26˚C whereas 37˚C resulted in the death of all PBMCs cultured [19]. In addition, the temperature of 20˚C, 27˚C and 28˚C have been recommended for the PBMCs studies in eel

**Table 2. Effect of blood anticoagulants on the peripheral blood mononuclear cells (PBMCs) isolation in *Varanus salvator*.**

| Items | Heparin | EDTA | P-value |
|---|---|---|---|
| | ————————PBMC, x10^5 cells/ml———————— | | |
| Day 0 | 2.00 | 2.00 | ND |
| Day 1 | 7.51[a] | 4.75[b] | 0.04 |
| Day 2 | 6.23[a] | 4.00[b] | 0.03 |
| Day 3 | 4.05[a] | 3.20[b] | 0.04 |
| Day 6 | 4.03[a] | 2.72[b] | 0.03 |
| Day 7 | 3.58[a] | 2.39[b] | 0.02 |
| Day 8 | 3.23[a] | 2.00[b] | 0.02 |
| Day 9 | 2.75[a] | 1.83[b] | 0.01 |
| Day 10 | 2.53[a] | 1.42[b] | 0.01 |

[a,b] Median values with different superscript letters within the same row were statistically significant (Wilcoxon test, $P < 0.05$).

ND = not determined.

**Table 3. Effect of culture temperatures on the peripheral blood mononuclear cells (PBMCs) isolation in *Varanus salvator*.**

| Items | 37˚C | 27˚C | *P*-value |
|---|---|---|---|
| | ————————PBMC, x10$^5$ cells/ml———————— | | |
| Day 0 | 2.00 | 2.00 | ND |
| Day 1 | 5.69 | 6.20 | 0.10 |
| Day 2 | 5.03 | 5.48 | 0.17 |
| Day 3 | 3.70 | 3.38 | 0.11 |
| Day 6 | 3.63[a] | 3.00[b] | 0.02 |
| Day 7 | 3.29[a] | 2.49[b] | 0.02 |
| Day 8 | 3.03[a] | 2.28[b] | 0.00 |
| Day 9 | 2.65[a] | 2.03[b] | 0.01 |
| Day 10 | 2.32[a] | 1.73[b] | 0.02 |

[a,b] Median values with different superscript letters within the same row were statistically significant (Wilcoxon test, $P < 0.05$).

ND = not determined.

[28], turtle [21] and catfish [28] respectively, which indicate the appropriate temperatures applied for ectothermic animals including fishes, amphibians and reptiles. This observation indicated that unlike mammals, the ectothermic animals have their own condition for peripheral blood cell activities. The evidence that Varanus' PBMCs can be treated in both temperature, especially in 37˚C, of this study was in accordance to their preferable living temperature, 27–37˚C [30]. Thus, this temperature range, 27˚C to 37˚C, might not only benefit their activities but play an important role on their immunity as well. This evidence can benefit laboratory resources and flexibility.

Considering how the PBMCs had changed during 10 days of experiment (Table 4), there was very interesting that PBMCs volume obtained from heparin and EDTA blood were

**Table 4. Effect of either blood anticoagulants or culture temperature on day culture effect of the peripheral blood mononuclear cells (PBMCs) isolation in *Varanus salvator*.**

| Items | Heparin | | EDTA | |
|---|---|---|---|---|
| | 37˚C | 27˚C | 37˚C | 27˚C |
| | ————————————PBMC, x10$^5$ cells/ml———————————— | | | |
| Day 0 | 2.00[d] | 2.00[d] | 2.00[de] | 2.00[cd] |
| Day 1 | 7.10[a] | 7.75[a] | 4.40[a] | 4.75[a] |
| Day 2 | 5.95[b] | 6.00[b] | 3.55[a] | 4.04[a] |
| Day 3 | 4.35[bc] | 3.95[c] | 3.15[b] | 3.00[b] |
| Day 6 | 4.50[bc] | 3.50[cd] | 2.93[bc] | 2.30[bc] |
| Day 7 | 4.00[c] | 2.90[cd] | 2.40[cd] | 2.18[bcd] |
| Day 8 | 3.85[cd] | 2.75[cd] | 2.19[cde] | 1.75[cd] |
| Day 9 | 3.25[cd] | 2.35[d] | 2.00[de] | 1.55[cd] |
| Day 10 | 2.95[cd] | 2.15[d] | 1.65[e] | 1.30[c] |
| *P*-value | 0.00 | 0.00 | 0.00 | 0.00 |

[a,b,c,d,e] Median values with different superscript letters within the same column were statistically significant (Friedman test, $P < 0.05$).

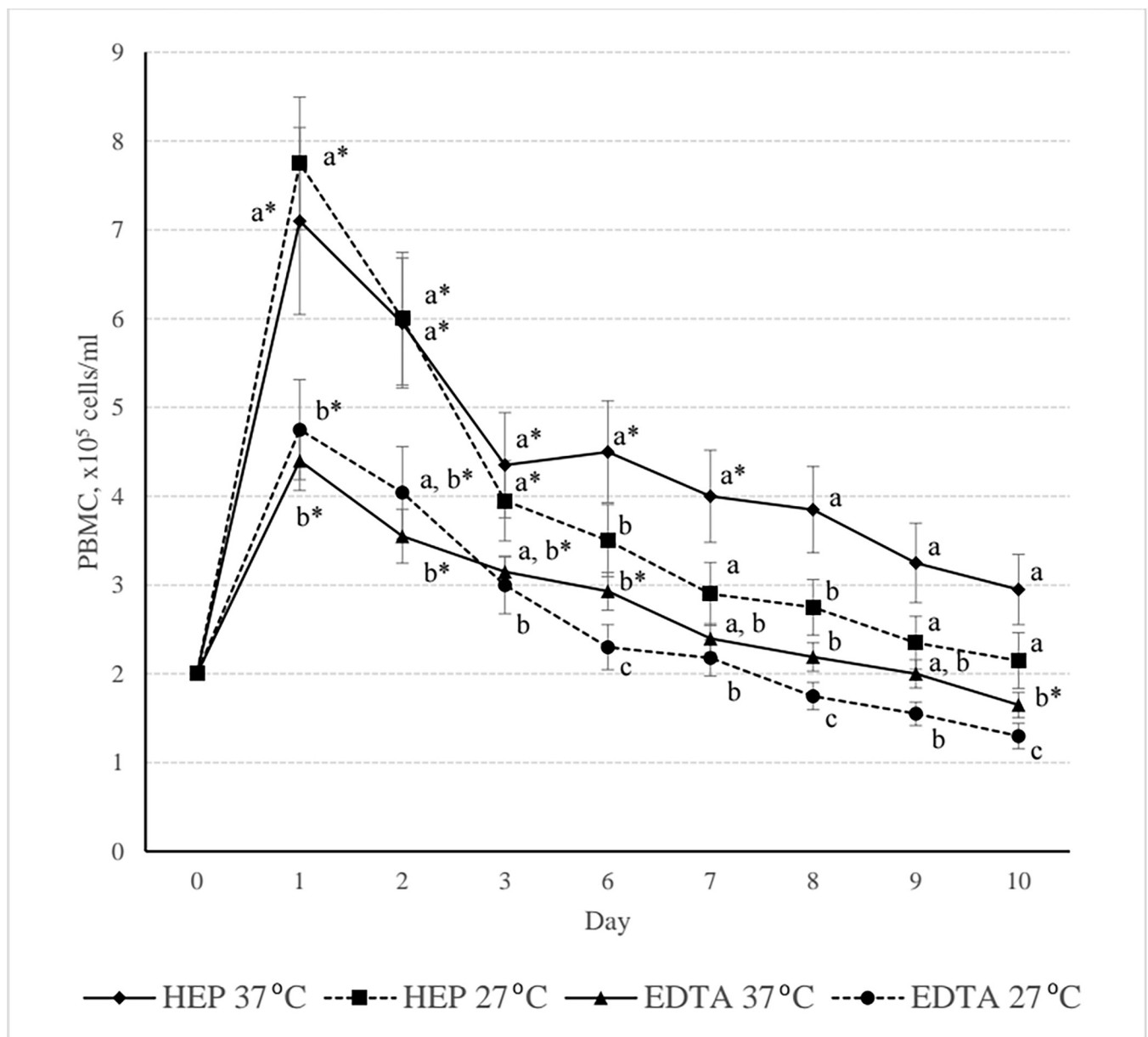

**Fig 1. Peripheral blood mononuclear cells (PBMCs) (median ± SE) of heparin blood (HEP) and EDTA blood (EDTA) from *Varanus Salvator* which was cultured at 37˚C and 27˚C for 10 days.** [a, b, c] Statistical significant of the effect of either blood anticoagulants or culture temperature on the PBMCs (Friedman test, $P < 0.05$) [*] Statistical significant when compared to day 0 (Friedman test, $P < 0.05$).

multiplied up to about 2 to 3 times on the first day of culture. The declination on the following days especially PBMC level from heparin blood was still presented beyond the baseline at day 10 (Fig 1). According to the heparin blood origin which cultured in 37˚C, PBMCs of day 1 was significantly higher than the others ($P < 0.05$), whereas day 2 was significantly higher than day 0, 7, 8, 9 and 10 ($P < 0.05$) (Table 4). The PBMCs isolated from heparin blood which culture under 27˚C was significantly higher in day 1 and 2 when compared with the others as well ($P < 0.05$) (Table 4). Considering to the PBMCs isolated from EDTA blood, under 37˚C culture, showed significantly higher in day 1, 2 and 3 when compared with day 0 and declined in following days ($P < 0.05$) (Table 4). For 27˚C culture, the significantly higher in day 1 and 2

were shown when compared with day 0 and declined in following days as well ($P < 0.05$) (Table 4). All remaining statistical differences are shown in Table 4. These results indicated that, depending on culture period, the *Varanus salvator*'s PBMCs isolated from both anticoagulants blood and cultured under both temperature factors were peak on day 1 and 2 and started to decline in following days. However, unlike other animals, Varanus' PBMCs spent 10 days to drop to the starting point (Fig 1), indicated an interested phenomenon which benefit for the further immunology investigation of the Varanus spp.

The unexpected results on first 3 days which similar to the stimulating cells pattern [23,39] should be discussed. Most animals especially the mammalian, the PBMCs culture should not proliferate without stimulus and had to decline during the incubation period, which should be resuspended in cell medium for their longer life in cell line protocol. In this experiment, the normal cow's PBMCs pattern had been performed in a parallel condition to represent a mammalian PBMC pattern without the stimulant (Fig 2). For this phenomenon, some explanation should be raised. One possibility is that Varanus always confront to a wide variety of microorganism, therefore, their PBMCs may remain in an active state which caused a highly proliferation during the experiment. Another possibility is the culture medium, Iscove is a high performance synthetic medium which suitable for rapid proliferating and high-density cell cultures in a 5% $CO_2$ atmosphere. In this case, Varanus PBMCs may be more susceptible to Iscoves than PBMCs from other animals. However, Iscove is considered to be the principal and most widely used medium in PBMCs cell culture processes. Therefore, depending on opportunity, the washing medium, RPMI-1640, may suitable to be applied for Varanus's PBMCs culture in some cases, however, the high proliferation of PBMCs in this study is valuable in preparing the cell line for further experiments. Finally, an evidence of Varanus's PBMC derived from heparin blood and culture in 37°C that showed the statistical significance of cell proliferation along 7 days (Fig 1) would encourage scientists for further PBMCs experiments which heparin should be the anticoagulant of choice of blood sampling and 37°C should be the preferably culture temperature for PBMCs preparation.

## The hematological characteristics

Blood samples were collected from a total of 49 Asian water monitors. Hematozoa were detected in the RBCs of 16 monitors (32.65%) without hemolytic specimens were observed. There were no significant differences between *Hepatozoon*-negative and *Hepatozoon*-positive Asian water monitors in every hematologic parameter which the data of each are shown as well as the combined (Table 5).

The peripheral blood cells of Asian water monitor were composed of erythrocytes, leukocytes and thrombocytes. Total number of erythrocytes were $1.19 \pm 0.04$ x $10^{12}$/L ($1.17$–$1.35$ x $10^{12}$/L) with PCV 32.00% $\pm$ 0.53% (31.69%-33.83%) from 49 samples (Table 5). In 49 samples of Asian water monitors, it contains WBC $2.41 \pm 0.13$ x $10^9$/L ($2.29$–$2.81$ x $10^9$/L). Along with absolute differential count, there are heterophils $0.92 \pm 0.02$ x $10^9$/L ($0.87$–$0.95$ x $10^9$/L), lymphocytes $1.17 \pm 0.01$ x $10^9$/L ($1.15$–$1.23$ x $10^9$/L), azurophils $0.40 \pm 0.01$ x $10^9$/L ($0.37$–$0.42$ x $10^9$/L), basophils $0.000 \pm 0.001$ x $10^9$/L ($0.004$–$0.011$ x $10^9$/L), and monocytes $0.027 \pm 0.002$ x $10^9$/L ($0.028$–$0.039$ x $10^9$/L) (Table 5).

By light microscopy, erythrocytes of Asian water monitor had similar appearance to those of other nonmammalian vertebrates. The cells were oval to elliptical in shape with round to oval concentric nuclei and abundant eosinophilic cytoplasm (Fig 3A). The width and length of mature erythrocytes were $8.52 \pm 0.20$ μm ($8.24$–$9.09$ μm) and $16.23 \pm 0.26$ μm ($15.57$–$16.67$ μm), respectively. A small amount of polychromatophilic or immature erythrocytes revealed in the peripheral blood. These cells tended to be more rounded than oval with a blue-

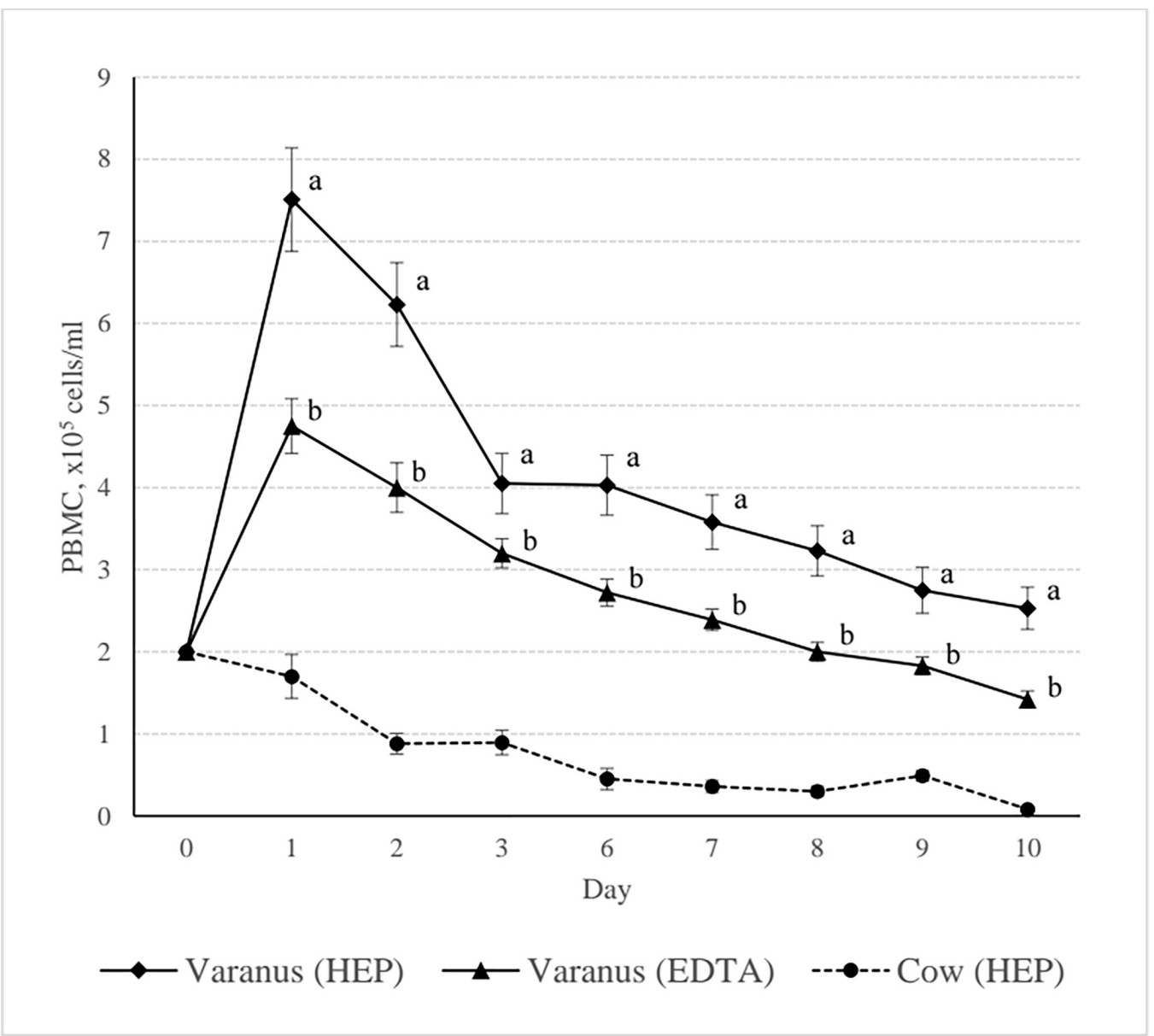

**Fig 2. Peripheral blood mononuclear cells (PBMCs) (median ± SE) of heparin blood (HEP) and EDTA blood (EDTA) from *Varanus Salvator* cultured for 10 days.** The PBMCs from cows (n = 10) represent the mammal's PBMC pattern. [a, b] Statistical significant of the effect of blood anticoagulants on the Varanus's PBMCs (Wilcoxon test, $P < 0.05$).

tinted cytoplasm, and larger, less dense chromatin in nuclei (Fig 3A). These cells were 10.29 ± 0.81 μm (7.68–11.66 μm) in width and 13.80 ± 0.71 μm (11.31–14.79 μm) in length.

Leukocytes of Asian water monitors were composed of granulocytes and agranulocytes. The granulocytic leukocytes were divided to heterophils and basophils according to their specific cytoplasmic granules. Whereas agranulocytes can be subdivided as azurophils, lymphocytes, and monocytes.

Heterophils were the most common granulocytes of Asian water monitors. They were spherical cells approximately 14.40 ± 0.79 μm (12.02–15.77 μm) in diameter with rounded eccentric nuclei. Their cytoplasm contained numerous elongated shaped granules that were bright eosinophilic and refractive in appearance (Fig 3B).

**Table 5. Hematological parameters of Asian water monitor comparison between the hepatozoon infection and non-infection (median ± SE and 95% confidence interval).**

| | *Hepatozoon*-positive (n = 16) | *Hepatozoon*-negative (n = 33) | Total (n = 49) |
|---|---|---|---|
| PCV (%) | 31.75 ± 0.52 (30.25–34.20) | 32.00 ± 0.53 (31.67–34.35) | 32.00 ± 0.53 (31.69–33.83) |
| RBC count (x10$^{12}$/L) | 1.18 ± 0.03 (1.09–1.34) | 1.23 ± 0.05 (1.17–1.40) | 1.19 ± 0.04 (1.17–1.35) |
| WBC count (x10$^9$/L) | 2.25 ± 0.11 (1.86–2.64) | 2.66 ± 0.16 (2.36–3.03) | 2.41 ± 0.13 (2.29–2.81) |
| Absolute differential count | | | |
| Heterophil (x10$^9$/L) | 0.77 ± 0.03 (0.72–0.86) | 0.99 ± 0.02 (0.94–1.01) | 0.92 ± 0.02 (0.87–0.95) |
| Azurophil (x10$^9$/L) | 0.33 ± 0.02 (0.29–0.40) | 0.42 ± 0.01 (0.40–0.44) | 0.40 ± 0.01 (0.37–0.42) |
| Basophil (x10$^9$/L) | 0.00 ± 0.002 (0.0002–0.01) | 0.00 ± 0.002 (0.004–0.01) | 0.00 ± 0.001 (0.004–0.011) |
| Lymphocyte (x10$^9$/L) | 1.10 ± 0.01 (1.03–1.11) | 1.26 ± 0.01 (1.21–1.29) | 1.17 ± 0.01 (1.15–1.23) |
| Monocyte (x10$^9$/L) | 0.02 ± 0.004 (0.020–0.04) | 0.027 ± 0.003 (0.028–0.04) | 0.027 ± 0.002 (0.028–0.039) |
| | *Hepatozoon*-positive (n = 16) | *Hepatozoon*- negative (n = 33) | Total (n = 49) |
| Relative differential count | | | |
| Heterophil (%) | 34.50 ± 1.48 (31.97–38.28) | 37.00 ± 0.67 (34.91–37.64) | 35.00 ± 0.66 (34.56–37.22) |
| Azurophil (%) | 15.00 ± 1.16 (13.16–18.09) | 15.50 ± 0.46 (14.68–16.56) | 15.00 ± 0.48 (14.65–16.60) |
| Basophil (%) | 0.00 ± 0.11 (0.01–0.52) | 0.00 ± 0.08 (0.17–0.52) | 0.00 ± 0.07 (0.18–0.46) |
| Lymphocyte (%) | 49.00 ± 0.85 (45.81–49.44) | 47.00 ± 0.69 (45.01–47.86) | 48.00 ± 0.54 (45.73–47.93) |
| Monocyte (%) | 1.00 ± 0.22 (0.90–1.85) | 1.00 ± 0.12 (1.06–1.56) | 1.00 ± 0.10 (1.11–1.55) |
| Plasma protein (g/L) | 82.50 ± 0.20 (77.55–92.94) | 84.75 ± 0.16 (80.88–88.75) | 82.50 ± 0.17 (81.47–88.44) |
| Thrombocytes (/100WBC) | 50.00 ± 1.65 (46.34–53.41) | 54.5 ± 1.28 (49.72–54.76) | 52.00 ± 1.02 (49.46–53.57) |

Basophils were smaller approximately half the size of heterophils. They were about 6.98 ± 0.02 μm (6.92–7.02 μm) in diameter. They had rounded nuclei and numerous rounded deeply stained purple cytoplasmic granules which often obscured the nuclei (Fig 3C).

Lymphocytes were the most prominent white blood cells in circulation. They were small round cell and ranged in size from small to large (Fig 3C–3E). Their diameter was 9.22 ± 0.47 μm (8.11–10.31 μm). They had spherical nuclei filled with dense chromatin. The cytoplasm was basophilic, scant, and may contain a few coarse azurophilic granule (Fig 3D).

Azurophils were large round cells, measured approximately 13.16 ± 0.48 μm (12.01–14.22 μm) in diameter. They had an eccentric round to oval to bilobed nucleus, basophilic cytoplasm and contained large numbers of fine cytoplasmic azurophilic granules (Fig 3F).

Monocytes were the largest leukocyte with the diameter about 14.79 ± 1.11 μm (11.30–17.52 μm). They had a round to amoeboid nucleus. The cytoplasm was abundant with a ground glass appearance (Fig 3G).

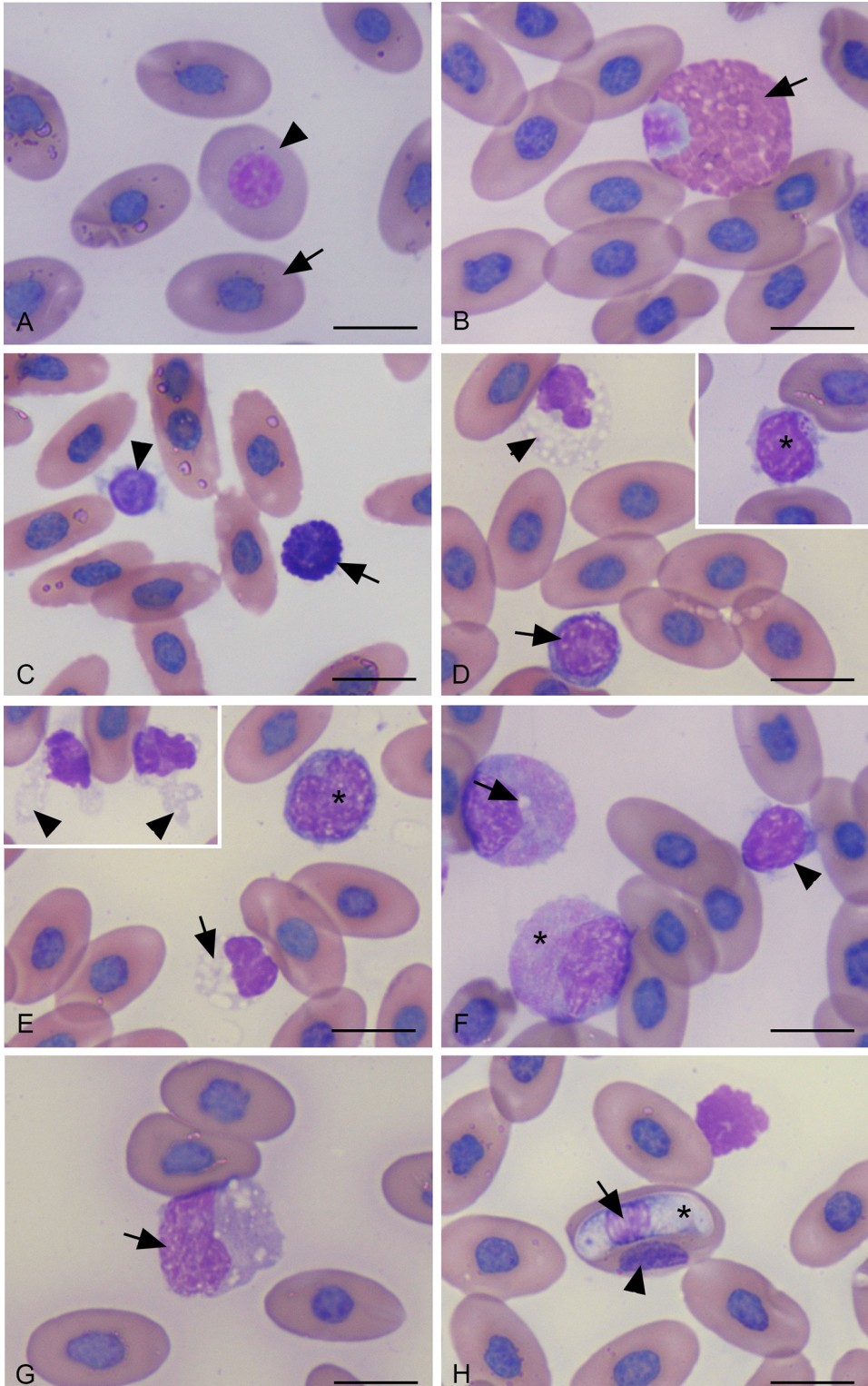

**Fig 3. Light micrographs of peripheral blood cells from Asian water monitor (Modified Wright Giemsa stained, horizontal bar: 10 micrometers).** (A) Erythrocyte (arrow), polychromatic erythrocyte (arrow head). (B) Heterophils shows eccentric rounded nucleus with abundant cytoplasmic eosinophilic fusiform granules (arrow). (C) Basophil (arrow), small lymphocyte (arrow head). (D) Thrombocyte (arrow head) with vacuolated cytoplasm, Small lymphocyte (arrow), Inset shows small lymphocyte with intracytoplasmic azurophilic granules (asterisk). (E) Large lymphocyte

(asterisk), thrombocyte with vacuolated cytoplasm (arrow), inset shows thrombocytes with cytoplasmic pseudopods (arrow heads). (F) Azurophils with fine azulophilic granules in cytoplasm (asterisk), the upper cells shows vacuole in cytoplasm (arrow), small lymphocyte (arrow head). (G) Monocyte (arrow) with abundant pale basophilic cytoplasm. (H) Erythrocyte with gamont of *Hepatozoon spp.* in cytoplasm, nucleus of erythrocyte (arrow head), nucleus (arrow) and cytoplasm (asterisk) of *Hepatozoon spp*.

Thrombocytes of Asian water monitor were variable in shape and may be rounded to oval shaped with moderate amount of colorless cytoplasm. They measured about 7.73 ± 0.54 µm (6.97–9.41µm) in diameter. Their nuclei were usually oval or segmented and filled with chromatin that more condense than those of the lymphocytes. A number of thrombocytes with cytoplasmic pseudopods were present (Fig 3D and 3E). The sizes of blood cells of Asian water monitors are given in Table 6.

In the present study also revealed 32.65% of hematozoa infected animals. Gametocytes of *Hepatozoon spp.* were identified within cytoplasm of infected erythrocytes. They were banana in shape, measured approximately 4.45 µm in width and 19.0 µm in length with a pale basophilic cytoplasm and central oval basophilic nuclei. The host cell nucleus was distorted and displaced to the opposite side (Fig 3H).

Hematology is an important tool in diagnosing and monitoring diseases or physiological disturbance in vertebrate animals. Studies of blood cells of Asian water monitor may provide information about the animals' health and therefore, may be used as a tool for diagnosis. In the present study, the peripheral blood of Asian water monitors was examined using quantitative and qualitative methods. The erythrogram parameters obtained in this study are in agree with those values reported for other lizard species, including crocodile monitors [40], Asian water monitors [10]. The red blood cell morphology of Asian water monitors was generally similar to previous reports in non-mammalian vertebrates [9,36,41]. Although hematozoa were diagnosed in the red blood cells of 16 monitors (32.65%), there were no significant differences between *Hepatozoon*-negative and *Hepatozoon*-positive animal in every hematologic parameter. Three genera of hemoparasites include *Hemogregarine*, *Hepatozoon* and *Karyolysus* species are commonly present in reptiles. They are nonpathogenic and transmitted by mites, ticks, mosquitoes and flies [9]. Polychromatophilic erythrocytes accounted for less than 1% of the total erythrocytes of this study. The immature erythrocytes are occasionally observed in peripheral blood of reptiles, particularly the very young animals [36]. The presence of

**Table 6. The sizes of peripheral blood cells of Asian water monitors (median ± SE and 95% confidence interval).**

|  | Width (µm) | Length (µm) | Diameter (µm) |
|---|---|---|---|
| Mature red blood cell | 8.52 ± 0.20 (8.24–9.09) | 16.23 ± 0.26 (15.57–16.67) | - |
| Polychromic red blood cell | 10.29 ± 0.81 (7.68–11.66) | 13.80 ± 0.71 (11.31–14.79) | - |
| Heterophil | - | - | 14.40 ± 0.79 (12.02–15.77) |
| Basophil | - | - | 6.98 ± 0.02 (6.92–7.02) |
| Lymphocyte | - | - | 9.22 ± 0.47 (8.11–10.31) |
| Azurophil | - | - | 13.16 ± 0.48 (12.01–14.22) |
| Monocyte | - | - | 14.79 ± 1.11 (11.30–17.52) |
| Thrombocytes | - | - | 7.73 ± 0.54 (6.97–9.41) |

increased polychromatic erythrocytes indicates active erythropoiesis [9]. The number of leukocytes observed in this study was lower than those of previous reported by Salakij et al. [10] but similar to the values of crocodile monitors, *Varanus salvadorii* [40] and Nile monitors, *Varanus niloticus* [42]. In amphibians and reptiles, the number of leukocytes in 1 milliliter of blood varies. It was reported that the variations in the number of leukocytes may related to gender, age, season, environmental and pathologic factors [35]. The morphological characteristic of granulocytes and agranulocytes obtained in this study were similar to those previous observed in other reptilian species [9,40] and the same specie water monitor [10]. Lymphocytes were the most abundant leukocytes in number, followed by heterophils, azurophils, monocytes, and basophils. These patterns show positive correlation with the previous reported for the crocodile monitors [40] and the same specie water monitor [10]. In contrast, azurophils were not present in the peripheral blood of Nile monitors [42]. The current study demonstrated cytoplasmic processes of thrombocytes which were not observed in thrombocytes in other varanid species [16,40,42]. A temporary arm -like projection of a eukaryotic cell membrane that are developed in the direction of movement usually present in phagocytic cells [43]. It has been reported the presence of cytoplasmic processes in thrombocytes of teleost fishes and amphibian with either phagocytosis or killing ability [44]. As with other vertebrates, reptilian thrombocytes besides playing a key role in hemostasis, are also involved in the defense mechanism through phagocytosis [9]. It was suggested that thrombocytes often been activated or ruptured during blood venipuncture or blood film preparation. Activated thrombocytes often accumulate and can form pseudopods or contain a few cytoplasmic vacuoles [9]. Activation is frequent in blood smears obtained from the caudal vein, and some of the changes noted in the cells on a smear may be caused by the activation of the clotting cascade [41]. This study provided the important information on the morphology of Asian water monitor thrombocytes that may related to their phagocytic activity. Further investigation is needed to clarify the possible roles of thrombocytes in phagocytosis.

## Conclusions

The preparation of the peripheral blood mononuclear cells (PBMCs) in this experiment reveal that the *Varanus salvator*'s peripheral blood should be drawn into blood container treated with heparin as the anticoagulant. The isolated Varanus's PBMCs can be cultured at 27˚C and 37˚C which 37˚C should be optimal in 5% $CO_2$ humidified atmosphere. The investigation, as well as, the hematological characteristics of Asian water monitor in this study can strengthen new information to encourage further researches on the cell immunology of the *Varanus salvator*.

## Acknowledgments

The research was supported by the Faculty of Veterinary Science, Mahidol University. The authors gratefully acknowledge Mr. Winan Wirana for forestry technical expertise at Khaoson Wildlife Breeding station and the department of national park wildlife and plant conservation for the sampling assistance and Asst. Prof. Dr. Surasak Chittakot for the statistical consultation.

## Author Contributions

**Conceptualization:** Jitkamol Thanasak.

**Data curation:** Jitkamol Thanasak, Tawewan Tansatit.

**Formal analysis:** Jitkamol Thanasak, Tawewan Tansatit.

**Investigation:** Jitkamol Thanasak, Tawewan Tansatit, Jarupha Taowan, Napawan Hirunwiroj, Sujit Chitthichanonte, Teetat Wongmack.

**Methodology:** Jitkamol Thanasak, Tawewan Tansatit, Jarupha Taowan, Napawan Hirunwiroj, Sujit Chitthichanonte, Teetat Wongmack.

**Project administration:** Jitkamol Thanasak.

**Resources:** Jitkamol Thanasak.

**Writing – original draft:** Jitkamol Thanasak, Tawewan Tansatit.

**Writing – review & editing:** Jitkamol Thanasak.

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
