## [Decision Letter · Decision Letter 0]

1 Mar 2022

PONE-D-21-29537The peripheral blood mononuclear cells preparation and the hematology of Varanus salvatorPLOS ONE

Dear Dr. Thanasak,

Thank you for submitting your manuscript to PLOS ONE. After careful consideration, we feel that it has merit but does not fully meet PLOS ONE’s publication criteria as it currently stands. Therefore, we invite you to submit a revised version of the manuscript that addresses the points raised during the review process.

We look forward to receiving your revised manuscript.

Kind regards,

Maria del Mar Ortega-Villaizan

Academic Editor

PLOS ONE

A clean copy of the edited manuscript (uploaded as the new *manuscript* file).

Additional Editor Comments:

Following reviewers comments, major revisions should be carried out before being accepted for publication.

Reviewers' comments:

Reviewer's Responses to Questions

**Comments to the Author**

1. Is the manuscript technically sound, and do the data support the conclusions?

Reviewer #1: Yes

Reviewer #2: Yes

2. Has the statistical analysis been performed appropriately and rigorously? 

Reviewer #1: No

Reviewer #2: Yes

3. Have the authors made all data underlying the findings in their manuscript fully available?

Reviewer #1: Yes

Reviewer #2: Yes

4. Is the manuscript presented in an intelligible fashion and written in standard English?

Reviewer #1: Yes

Reviewer #2: Yes

5. Review Comments to the Author

Reviewer #1: The ms is aimed to evaluate the proper anticoagulants for the peripheral blood mononuclear cells (PBMCs) isolation and to analyse the effect of culture temperature on cell proliferation in the Varanus salvator’s. The ms is well written and the aims clearly explained and analysed. The main weakness I found rely with sample size and statistical analyses.

1)Males and females have been cumulated, but sex based differences are commonly found in lizards. So I suggest to separate sexes in all the analyes

2)Statistical methods are very poorly described, and in some cases are not fully appropriated For example, The simple ANOVA is not appropriate when comparing means with repeated sampling wihin individuals as, for example, in the analysis of the effect of temperature on cell proliferation (see Sacchi R, Mangiacotti M, Scali S, Coladonato AJ, Pitoni S, Falaschi M, et al. (2020) Statistical methodology for the evaluation of leukocyte data in wild reptile populations: A case study with the common wall lizard (Podarcis muralis). PLoS ONE 15(8): e0237992.). The data reported in figg 1 and fig 2 should be statistically processed by a LMM in which the number of cells is the dependent variable; the day, the treatments (temperature and anticoagulant) and their interactions are the predictors, while the individual is the random effect to account for repeated measuring.

3)Further, body size has been totally omitted in the analysis, but we know that size (i.e. age) could affect cell growth.

4)I believe that some of the differences between this and previous studies pointed out in several part of the ms might be due to the not appropriate statistical treatment of data

5)A further problem rely with the reference intervals, as reported in table 1. What do errors represent? I suppose standard deviations, but 95% CI intervals shoud be more appropriate. (see Dickinson VM, Jarchow JL, Trueblood MH. Hematology and plasma biochemistry reference range values for free-ranging desert tortoises in Arizona. J Wildl Dis. 2002; 38:143–153.)

A final point is the ripercussion in conservation of the results of this study (line 101). Since only captive individuals have been considered, reference interval and physiological responses here analysed cannot directly extrapolated to wild populations, since captivity has relevant effects on blood parameters. In wild populations the variability of blood values is much higher than that in captive individulas (see Sacchi R, Mangiacotti M, Scali S, Coladonato AJ, Pitoni S, Falaschi M, et al. (2020) Statistical methodology for the evaluation of leukocyte data in wild reptile populations: A case study with the common wall lizard (Podarcis muralis). PLoS ONE 15(8): e0237992.). I think this point should be accounted when discussing the relevance of the results of the study. I think that main ripercussion of the results concenrca pets, captive animals and veterinary procedures, rather than species conservation

Reviewer #2: The immunological characteristics of the Varanus Salvator are very unknown. To evaluate the hematological characteristics and discriminate some practical conditions to culture, immune cells (PBMCs) of Varanus Salvator have an interest and it is a good tool to study. The descriptive study is well executed and presented.

Some questions have to be resolved:

Material and Methods

Lines 118-119. All blood samples, both the obtained with heparin and the obtained with EDTA were treated with heparin? Can you study the effects of EDTA if all samples were treated with heparin?

Lines 127-128. You indicate that the PBMCs lifespan were monitored at day 1, 2, 3, 4, 5, 6, 7, 8, 9, and 10 were by Trypan blue method; however, in the figures you indicate the cell number at 0 days. How you determine the cell counts at 0 days? Perhaps the high increase in cells at day 1 respect to day 0 could be results of different measurement techniques of cell count these days.

Results and Discussion

Can you explain the proliferation of PBMCs in culture? PBMCs increase about three times the first culture day. Can PBMCs proliferate in culture?

6. PLOS authors have the option to publish the peer review history of their article (what does this mean?). If published, this will include your full peer review and any attached files.

Reviewer #1: No

Reviewer #2: No

---

## [Author Response · Author response to Decision Letter 0]

15 Apr 2022

Dear reviewers

According to some points raised by the academic reviewers. Please see the responses to each point in the file labeled 'Response to Reviewers'. Many thanks for all the helpful points. 

Your sincerely

The authors of manuscript 

The peripheral blood mononuclear cells preparation and the hematology of Varanus salvator

---

## [Decision Letter · Decision Letter 1]

16 May 2022

The peripheral blood mononuclear cells preparation and the hematology of Varanus salvator

PONE-D-21-29537R1

Dear Dr. Thanasak,

We’re pleased to inform you that your manuscript has been judged scientifically suitable for publication and will be formally accepted for publication once it meets all outstanding technical requirements.

Kind regards,

Maria del Mar Ortega-Villaizan

Academic Editor

PLOS ONE

Additional Editor Comments (optional):

The manuscript considerably improved following reviewers indications and is ready for publication

Reviewers' comments:

Reviewer's Responses to Questions

**Comments to the Author**

1. If the authors have adequately addressed your comments raised in a previous round of review and you feel that this manuscript is now acceptable for publication, you may indicate that here to bypass the “Comments to the Author” section, enter your conflict of interest statement in the “Confidential to Editor” section, and submit your "Accept" recommendation.

Reviewer #2: All comments have been addressed

2. Is the manuscript technically sound, and do the data support the conclusions?

Reviewer #2: Yes

3. Has the statistical analysis been performed appropriately and rigorously? 

Reviewer #2: Yes

4. Have the authors made all data underlying the findings in their manuscript fully available?

Reviewer #2: Yes

5. Is the manuscript presented in an intelligible fashion and written in standard English?

Reviewer #2: Yes

6. Review Comments to the Author

Reviewer #2: The changes introduced into the manuscript and the explanations resolve all the issues raised. The explanation of the proliferation of PBMCs in culture could be reasonable.

7. PLOS authors have the option to publish the peer review history of their article (what does this mean?). If published, this will include your full peer review and any attached files.

Reviewer #2: No

---

## [Editor Report · Acceptance letter]

14 Jul 2022

PONE-D-21-29537R1 

The peripheral blood mononuclear cells preparation and the hematology of *Varanus salvator*

Dear Dr. Thanasak:

I'm pleased to inform you that your manuscript has been deemed suitable for publication in PLOS ONE. Congratulations! Your manuscript is now with our production department. 

Kind regards, 

on behalf of

Dr. Maria del Mar Ortega-Villaizan 

Academic Editor

PLOS ONE